# Alternative Methods as Tools for Obesity Research: In Vitro and In Silico Approaches

**DOI:** 10.3390/life13010108

**Published:** 2022-12-30

**Authors:** Juliana Helena Pamplona, Bernardo Zoehler, Patrícia Shigunov, María Julia Barisón, Vanessa Rossini Severo, Natalie Mayara Erich, Annanda Lyra Ribeiro, Cintia Delai da Silva Horinouchi, Andréia Akemi Suzukawa, Anny Waloski Robert, Ana Paula Ressetti Abud, Alessandra Melo de Aguiar

**Affiliations:** 1Laboratório de Biologia Básica de Células-Tronco, Instituto Carlos Chagas, FIOCRUZ Paraná, Curitiba 81350-010, PR, Brazil; 2Laboratório de Pesquisa em Apicomplexa, Instituto Carlos Chagas, FIOCRUZ Paraná, Curitiba 81350-010, PR, Brazil; 3Rede de Plataformas Tecnológicas FIOCRUZ—Bioensaios em Métodos Alternativos em Citotoxicidade, Instituto Carlos Chagas, FIOCRUZ Paraná, Curitiba 81350-010, PR, Brazil

**Keywords:** obesity, adipogenesis, mesenchymal stem cells, human adipose-derived stem cells, in vitro alternative methods, in silico alternative methods, 3D culture, microfluidics

## Abstract

The study of adipogenesis is essential for understanding and treating obesity, a multifactorial problem related to body fat accumulation that leads to several life-threatening diseases, becoming one of the most critical public health problems worldwide. In this review, we propose to provide the highlights of the adipogenesis study based on in vitro differentiation of human mesenchymal stem cells (hMSCs). We list in silico methods, such as molecular docking for identification of molecular targets, and in vitro approaches, from 2D, more straightforward and applied for screening large libraries of substances, to more representative physiological models, such as 3D and bioprinting models. We also describe the development of physiological models based on microfluidic systems applied to investigate adipogenesis in vitro. We intend to identify the main alternative models for adipogenesis evaluation, contributing to the direction of preclinical research in obesity. Future directions indicate the association of in silico and in vitro techniques to bring a clear picture of alternative methods based on adipogenesis as a tool for obesity research.

## 1. Introduction

According to the World Health Organization, one billion and nine hundred million adults worldwide were overweight in 2016; of these, 650 million were obese, corresponding to 39% and 13% of the world population, respectively [1]. Overweight and obesity were considered a problem only in high-income countries, however they are now on the alarming rise in low- and middle-income countries, becoming the most significant causes of ill health [1].

Morbidity and mortality associated with obesity have been related to a reduced quality of life, and it impacts life expectancy, as reviewed [2]. Comorbidities such as metabolic syndrome, insulin resistance, diabetes, cardiovascular diseases, musculoskeletal disorders, and some types of cancer are associated with obesity [1,3]. In general terms, obesity and overweight may be related to an energy imbalance between calories consumed and calories expended, which results in enhanced fat storage [1]. The overview of obesity in the world is shown in Figure 1. Some other factors are pointed out as obesity causes, including eating patterns, physical activity levels, sleep routine, genetics, medicine usage, and social determinants of health [1,3,4].

In spite of the efforts to find newer anti-obesity medication (AOM), most of them fail to provide consistent and sustainable weight loss at tolerable doses. Because of this, the research of AOM with fewer side effects and better results has been under the spotlight, as reviewed [5]. As it is known, clinical trials can be conducted after well-established knowledge based on nonclinical studies. Although animal models have been applied for the understanding of obesity causes and investigating treatments (reviewed by [6,7,8]), there are gaps and limitations because no animal model can completely reproduce the complexity of the human condition [8]. 

Even though overweight and obesity are multifactorial conditions [3,4], the biological role of adipocytes and adipogenesis may be addressed as a tool to understand the cause and the search for therapeutic approaches. Briefly, at the cellular level, obesity is related to an increase in the size (hypertrophy) and number (hyperplasia) of adipocytes from the white adipose tissue (WAT). In addition to adipocytes, mesenchymal stem cells (MSCs), adipocyte precursors, endothelial and immune cells are also components of the WAT. MSCs can undergo differentiation to give rise to new adipocytes that can store triglycerides in organelles called lipid droplets [9]. In this scenario, a better understanding of adipose stem cells and adipogenesis could lead to new strategies for studying obesity. Since adipogenesis is a complex process involving several molecular mechanisms, adipogenesis regulators, among others, have been studied as potential AOM as reviewed [5,9]. The first part of this review discusses how gene expression and metabolism are regulated during the adipogenesis of hMSCs and lipolysis. The second part describes the main in silico and in vitro approaches, including 2D and 3D cell cultures, bioprinting, and microphysiological systems applied to adipogenesis research. Limitations and future directions of these methodologies are also discussed to provide a comprehensive understanding of the usefulness of hMSCs as models for obesity research.

## 2. Gene Expression in Adipogenesis and Lipolysis

Human mesenchymal stem cells (hMSCs) are adult stem cells and can be isolated from numerous sources, such as bone marrow and adipose tissue, as reviewed [10]. These cells can be cultured in vitro and stimulated to undergo differentiation towards adipocytes by using specific inducers, as will be discussed in the following sections. After the preadipocyte commitment, terminal differentiation gives rise to adipocytes containing intracellular lipid droplets, as reviewed [11]. The route from MSCs to the formation of mature adipocytes is a well-coordinated process that requires the expression of several genes, their transcriptional factors, and signaling intermediates from numerous pathways (reviewed by [9]). But does in vitro adipogenesis recapitulate what happens in vivo? How to associate these in vitro processes with a multifactorial condition such as obesity? Even though the direct comparison between in vitro and in vivo data would be an intuitive approach, it is not trivial since adipose tissue expresses more than 8200 genes, including more than 120 receptors and 74 transcription factors [12].

Interestingly, gene expression and proteomic and metabolomic profiling of adipose tissue can also diverge depending on the source of fat depots in the human body [13,14]. Single-cell sequencing technologies have enabled a refined investigation of the transcriptomic profile of complex tissues such as adipose tissue. Recently, Norreen-Thorsen and colleagues established a human adipose tissue cell-type transcriptome atlas using unfractionated human adipose tissue RNA-seq data (visceral and subcutaneous) with over 2300 cell-type-enriched transcripts [15]. 

Noteworthy, sex-related differences in human adipose tissue may also be considered during gene expression analysis. For instance, FADS1 and MAP1B are genes implicated in oxidative phosphorylation and adipogenesis and can be influenced by sex differences [16]. A panel of male-only cell-type-enriched genes may also be considered a potential bias during data analysis [15]. Altogether, these findings highlight the diversity of gene expression and regulation related to adipose tissue in vivo.

In vitro adipogenesis has been extensively used to simulate the differentiation process because it allows controlling many variables, such as MSCs source, donor age, sex, confluence, passage, concentration, and types of differentiation inducers, differentiation induction time, among others. Different studies have been performed to evaluate gene expression profiles (reviewed by [17,18]) and membrane proteomic analysis [19] during in vitro adipogenesis. The evaluation of the upregulated and downregulated genes during in vitro adipogenesis can be used as an indirect method in searching for molecules that interfere with the process. For example, Foley and colleagues used the gene expression of key regulators of adipogenesis, such as the transcription factor CCAAT/enhancer binding protein alpha (CEBPA), the Peroxisome Proliferator-Activated Receptor Gamma (PPARγ), Perilipin 1 (PLIN1), and the fatty acid binding protein 4 (FABP4), to compose a multi-endpoint approach for drug screening. The authors screened 60 chemical compounds using an in vitro adipogenesis assay with human adipose-derived stem cells. They concluded that the determination of these and other key phenotypic endpoints represent an exciting approach to evaluating the effect of candidate molecules on adipogenic differentiation [20].

Different studies reported the upregulation of PPARγ and CEBPA during preadipocyte commitment and terminal differentiation (reviewed by [17]). Both promote differentiation by activating adipose-specific gene expression and maintaining each other’s expression at high levels. PPARγ is required to differentiate adipose tissue in vivo and in vitro [21]. Its activation stimulates MSCs to differentiate towards adipocytes and is responsible for synthesizing fat acids and triglycerides in a lipogenesis process [22]. Obesity and nutritional factors influence the expression of PPARγ in human adipocytes [23] and the existence of conditions, such as type two diabetes, once the expression of PPARγ and CEBPA is related negatively to fat cell size in non-obese men with type two diabetes [24]. Overweight/obese subjects presented with lower expression of adipogenesis markers, insulin signaling, and angiogenesis markers in the subcutaneous adipose tissue than normal-weight individuals [25]. Implications of obesity in the function of hMSCs have also been addressed. The lower expression of PPARγ but not CEBPA was also reported in hMSCs harvested from obese in comparison with non-obese subjects [26], which also reported a similar differentiation potential of hMSCs from both groups, but higher expression of senescence markers in hMSCs from obese subjects. 

Single-cell transcriptome demonstrated the existence of four additional adipocyte subtypes from human mesenchymal progenitors in vitro [27]. The analysis of the PPARγ expression profile of each cell subtype revealed differences between the four categorized groups. These findings are helpful in the identification of drugs/molecules that may have a specific role in each subtype, such as Forskolin, which strongly induced the expression of thermogenic genes UCP1, DIO2, and CIDEC, corresponding to thermogenic beige/brite adipocytes [27]. The heterogeneity of MSC and variable adipogenesis ratio were empirically determined, and currently, through single-cell sequencing, we can finely define these populations according to their gene expression regulation.

Lipolysis is the sequential hydrolysis of triacylglycerol (TAG) stored in cell lipid droplets [28]. Interestingly, it was found that CEBPA activates the transcription of a lipase during the differentiation of 3T3-L1 murine preadipocytes into adipocytes [29]. A low expression of classical lipases in human abdominal subcutaneous adipose tissue is accompanied by an increased expression of autophagy-related genes (ATGs) in overweight/obese subjects, which seems to be mainly related to glucose tolerance [30]. Some naturally derived compounds have been implicated in lipolysis and the reduction of adipogenic differentiation [31,32]. 

As will be discussed in the following sections, gene and protein expression studies may be applied in combination with in vitro and in silico approaches to investigate adipogenesis.

## 3. Metabolic Features as Endpoints to Evaluate Adipogenesis and Lipolysis in MSCs Models

The metabolic conditions of the cell are essential to maintain cell identity and in the decisions to proliferate or differentiate into different cell types [33,34]. Regarding their energy metabolism, it has already been described that stem cells, pluripotent and derived from adult tissues, vary in relation to differentiated cells. MSCs reside in hypoxic microenvironments (1–9% O_2_) [35,36] and appear to depend mainly on glycolysis for ATP production. Higher glycolytic enzymes and lactate production levels were detected in undifferentiated MSCs [37]. In contrast, MSCs induced by adipogenic differentiation adopt a metabolism based on mitochondrial oxygen consumption and reactive oxygen species (ROS) generation [34]. Different studies demonstrated that the increase in mitochondrial metabolism is essential for adipogenesis promotion and not a mere consequence of differentiation induction [38,39]. Our group investigated the first steps of adipogenic differentiation of MSC from a molecular and biochemical point of view. Commitment to adipocytes occurs three days after in vitro adipogenesis induction of MSCs, and gene expression analysis showed an energy metabolism profile different from undifferentiated MSCs [40]. An increase in ROS production was observed after three days of differentiation and could be involved in adipocyte commitment. After seven days, higher oxygen consumption and mitochondrial activity were observed, indicating a transition to oxidative metabolism [41]. Reactive oxygen species are implicated in adipogenic and osteogenic MSCs differentiation. It was proposed that ROS could interact with different signaling pathways that regulate MSCs differentiation, such as Hedgehog, Wnt, and FOXO [42].

Interestingly, Tormos and colleagues demonstrated that ROS produced in mitochondria could induce the expression of PPARγ [38]. This factor and others, such as CEBPA and sterol regulatory element binding transcription factor 1 (SREBP1c), not only regulate and determine the final stages of adipogenesis but also coordinate the expression of several genes involved in lipid metabolism, regulating lipogenesis and lipolysis pathways [43,44]. 

During adipogenesis, lipids are synthesized, mainly in the form of TAG, which accumulates in lipids droplets in the cytoplasm of adipocytes. The source of fatty acids (FAs) for TAG synthesis can be exogenous by the uptake of FAs and subsequent esterification to glycerol-3-phosphate or synthesized by the cells in a process denominated de novo lipogenesis (DNL). In this pathway, FAs are synthesized from glucose. After its uptake by glucose transporters as solute carrier family 2 (facilitated glucose transporter), member 4 (Glut4) and its metabolization through glycolysis and the Krebs cycle, the citrate produced from glucose is converted to acetyl-CoA by the ATP citrate lyase (ACL) and subsequently metabolized by acetyl-CoA carboxylase (ACC1) to produce malonyl-CoA. This intermediate is used by fatty acid synthases (FASN). This key enzyme in DNL condenses acetyl-CoA molecules to malonyl-CoA, producing longer FAs such as palmitoyl-CoA [45]. Interestingly, the acetyl-CoA produced by ACL is used along differentiation to histone acetylation in response to glucose availability. It was described that this epigenetic regulation is required for the expression of critical metabolic enzymes such as Glut4, hexokinase 2 (HK2), phosphofructokinase-1 (PFK1), and lactate dehydrogenase A (LDHA) [46], showing how gene expression regulation is also related to metabolism.

On the other hand, under starving conditions, TAGs are hydrolyzed by sequential lipases to FAs and glycerol in lipolysis. First, the adipose triglyceride lipase (ATGL) hydrolyzes TAGs to diacylglycerol, which is converted to monoacylglycerol by the enzyme hormone-sensitive lipase (HSL). Finally, a monoacylglycerol lipase generates glycerol and fatty acid. Fatty acids are oxidized within mitochondria via the β-oxidation pathway, supplying energy to the cell. First, FAs are converted to acyl-CoAs by an acyl-CoA synthetase activity. Then, a carnitine palmitoyltransferase 1A (CPT1A) located in the outer mitochondrial membrane converts the acyl-CoAs to acyl-carnitine to be transported to mitochondria by the carnitine shuttle. A CTP2 in the inner membrane reconverts acyl-carnitine to acyl-CoA, which enters the β-oxidation in the mitochondrial matrix. Sequentially, acetyl-CoA moieties are released in the cycle, shortening the acyl-CoAs and producing NADH and FADH_2_ as reducing equivalents [45,47,48]. Another relevant protein involved in lipid metabolism is FABP4, one of the most abundant proteins in adipocytes [49], which is responsible for the transport and storage of FAs, maintaining homeostasis in the cells [49,50]. FABP4 regulates adipogenesis by interaction with PPARγ [51] while participating in lipolysis through its interaction with HSL [52]. 

These metabolic features are frequently used as endpoints in AOM prospective research to evaluate drug effects during and post-adipogenesis. The analysis of transcript and protein expression levels of adipogenesis and lipolysis-related genes, as well as biochemical analysis based on the detection of TAG (adipogenesis), glycerol levels (lipolysis), or specific enzymatic activities, are also used to evaluate the effect of drugs with anti-obesity potential [53]. Figure 2 shows some of the genes whose expression or activity are commonly employed as endpoints, and examples of their use in the literature are described below. Using 3T3-L1 cells, Bu and colleagues showed that bilobalide, a compound derived from Gingko Biloba leaf, interferes with lipid metabolism, decreasing adipogenesis and lipid accumulation while increasing the expression of proteins involved in lipolysis. Specifically, the master regulators PPARγ and CEBPA; DNL-related genes such as ACC1, FASN, and GLUT4 were down-regulated, while lipolytic genes such as ATGL and HSL, and CPT1A were upregulated in a mechanism involving the activation of the AMPK signaling pathway [32]. On the other hand, the citrus bergamia extract decreased adipogenesis and increased lipolysis by modulation of PPARγ levels in hMSCs [31]. A study using MSCs derived from adipose tissue showed that celastrol, a bioactive compound from *Tripterygium wilfordi*, can inhibit adipogenesis in a dose, time of administration, and duration-dependent manner. Authors detected a lower lipid accumulation and a decreased expression of FABP4 as indicators of anti-adipogenic effect [54]. Resveratrol, a natural polyphenolic molecule with anti-obesity activity [55,56], inhibits the differentiation of human visceral preadipocytes (HPA-v) by decreasing fat accumulation in lipid droplets. Accordingly, expression of the key genes FABP4, ACC1, and FASN was down-regulated in cells treated with resveratrol in a dose-dependent manner [57]. White and brown adipocytes play different functions in the adipose tissue: while white adipocytes are involved in the storage and energy production from TAGs, brown adipocytes function as body temperature regulators, using stored lipids as fuel to heat production, via the uncoupling of mitochondrial respiration by the uncoupling protein 1 (UPC1). In this way, it was suggested that brown adipocytes could be relevant in anti-obesity treatments [58]. In a recent study, Senamontree and colleagues showed that betulinic acid (BA), a natural pentacyclic triterpenoid, reduces white adipogenesis while increasing brown adipogenesis of human MSCs. As metabolic endpoints, the authors describe a decrease in glycerol-3-phosphate dehydrogenase activity, a key enzyme in lipid biosynthesis, and an increase in glucose uptake. Moreover, BA induced the formation of smaller lipid droplets and a higher expression of UPC1, features of brown adipocytes, suggesting that betulinic acid could play a role in metabolism regulation and brown adipogenesis [59]. 

## 4. Alternative Methods to Animal Testing in Adipogenesis

Nonclinical studies have contributed to understanding the basic parameters that regulate energy balance, nutritional strategies, and the screening of AOM, among others. However, despite all the efforts, no animal model of obesity can reproduce the human condition, and there are many gaps and limitations of those models for obesity (reviewed by [6,7,8]).

Alternative approaches should be considered to study multifactorial conditions such as obesity. Non-animal tests have been applied in several countries for scientific purposes [60]. The main current alternative methods to the use of animals are the in silico and in vitro assays, which can be used cost-effectively to address the 3Rs principles related to reducing, replacing, or refining animal use, as first mentioned by Russell and Burch, published in 1959 as the 3Rs strategy [61]. 

In the following sections, we will discuss the main alternative methods for adipogenesis research and their impacts on obesity studies.

### 4.1. In Silico Methods May Provide the Basis for Elucidating the Mechanisms Underlying Adipogenesis

In recent years, bioinformatics has been widely applied in researching several diseases and biological processes. Full use of these big data sets has provided brilliant value for life science research [62]. Applying in silico methodologies can help to identify key factors that act in the adipogenesis process and reveal new therapeutic targets for associated metabolic syndromes such as obesity, type 2 diabetes, and other pathologies related to fat metabolism [18]. As adipogenesis is a highly controlled biological process, evaluating gene/protein expression profiles may allow us to determine the main actors responsible for regulating the entire process [63].

New insights into the adipogenesis process can be gained through in silico analyses. Gene enrichment analysis, such as gene ontology (G.O.), may provide new information and unveil promising genes for further analysis of adipogenic differentiation [64]. Upregulated genes during differentiation have been associated with collagen fibril organization, redox process, and cell receptor signaling pathway [63,64]. Importantly, such analyses allowed us to identify genes whose role in adipogenesis was previously unknown, such as ACSL1, S1PR3, ZBTB16, and GPC3. These may be closely associated with early-stage adipogenesis [64,65].

Additionally, the search for interactions between signaling molecules and biomarkers allows the identification of pathways closely related to adipogenic differentiation and its key factors. Analyses of full-genome microarray data using databases such as the Kyoto Encyclopedia of Genes and Genomes (KEGG) revealed the involvement of DUSP14, NTF3, FGF14, FGF7 genes, and CD14 molecules in the AMPK signaling pathway during in vitro adipogenesis of hMSCs [64]. Pathway analysis based on KEGG using five transcriptome datasets from differentiated 3T3-L1 adipocytes revealed upregulated genes in metabolic pathways in the proliferator-activated receptor signaling pathway and in regulating lipolysis in adipocytes, such as fat digestion and absorption, adipocytokine signaling pathway, and insulin signaling pathway. Some of the genes identified are critical in the transcriptional regulation of adipogenesis (PPARγ), insulin-sensitive glucose transport (SLC2A4), fatty acid transport (FABP4, CD36), triacylglycerol synthesis (DGAT1, DGAT2, ACSL1), lipolysis (LIPE, PNPLA2, ABHD5), and adipocyte endocrine functions (ADIPOQ, CFD, RETN) [63].

In silico analyses are also interesting in identifying potential key genes involved in the regulation of adipogenic differentiation. Genes/proteins enriched in adipogenesis pathways have been shown to participate in the in vitro differentiation process of different cell types, including MSCs [65]. Moreover, as adipogenesis is a fundamental process for fat metabolism, it is urgent to develop an approach to support clinical management and avoid complications related to metabolic imbalance, as in the case of obesity. For example, Yu and collaborators (2022) used a microarray dataset comprising obese and healthy patients to identify differentially expressed genes and enable targeted drug screening. The authors used a drug-gene interaction database that pointed out potential candidates for anti-obesity therapy, such as lipid-modifying drugs like gemfibrozil, fluvastatin, pravastatin, and atorvastatin [66].

Another tool, molecular docking, allows the prediction of likely sites of interactions and bonds between proteins or proteins and small molecules [67,68,69]. Several studies have used molecular docking in the adipogenesis context to search for therapeutic alternatives for treating obesity. For example, this technique was used to determine the interaction positions of curcuminoids with PPARγ, CEBPA, and acetyl-CoA carboxylase-lipogenic enzyme (ACC) since in vitro adipogenesis of 3T3-L1 cells were differently affected depending on the derivative. Molecular docking data showed that despite the structural similarities of the curcuminoids, they had different interaction positions with ligands and could possibly inhibit the expression of CEBPA and ACC [70].

Further, molecular docking studies were performed to investigate interactions of the pharmacological agents’ arbutin (ARB), purpurin (PUR), quercetin (QR), and pterostilbene (PTS) with the beta-ketoacyl reductase (KR) and thioesterase (TE) domains of fatty acid synthase (FAS) enzyme. All compounds displayed significant binding interactions with KR and TE domains of the FAS enzyme, supporting in vitro studies that significantly decreased the adipocyte differentiation and stimulatory effect on fatty acid uptake in 3T3-L1 adipocytes [71].

Another highly effective approach is the in silico drug repurposing technique, where an established drug molecule with an approved indication is tested for a newer pharmacological action [72,73]. Studies using this technique allowed the identification of floxacillin as a safe and effective drug candidate with anti-obesity activity against human FABP4, where a virtual screening was performed using a library of ligands with various FDA-approved drugs [72].

To perform docking studies, high-resolution X-rays, NMR, or homology-modeled structures are required. Several structure databases, such as the Protein Data Bank (PDB) [74], and ligands databases, such as the PubChem database [75], are freely accessible. Docking methods are modeled by matching ligand atoms at the receptor binding site. The final complex formation is optimized by steric, hydrophobic, and electrostatic complementarity [69]. Several docking software is available such as AutoDock Vina, with an optimized algorithm using multithreading speeds molecular docking and virtual screening with improved accuracy of the binding mode prediction [76].

The suggested pipeline described in Figure 3 outlines the complete process of in silico analysis of adipogenesis.

It is clear that in silico methodologies have limitations, and the bioinformatic tools are to predict only the potential genes and drugs central in the entire metabolic pathway. The results must be researched and validated with other methodologies, such as cell culture. Furthermore, to better understand the functions of the predicted key factors, large-scale validation studies and the elucidation of the molecular processes of adipogenesis should be carried out. Even so, the data obtained may provide new information and serve as a basis for future studies to elucidate the mechanism underlying adipogenic differentiation.

### 4.2. Inducers for Adipogenesis In Vitro

The increasing prevalence of obesity among the population has led to a growing interest in understanding the intricate and well-orchestrated mechanisms governing adipogenesis. In the past few years, this complexity has been majorly addressed through the employment of in vitro models that aim to recapitulate adipocyte commitment and differentiation in a precise and cost-effective way [11]. 

In order to generate adipocytes in vitro, a set of adipogenic inducers can be used to trigger the expression of several transcription factors known to regulate adipogenesis in vitro and the culture media combination of those inducers [77]. One of the most used adipogenic cocktails available is composed of dexamethasone (DEX), isobutyl-1-methylxanthine (IBMX), indomethacin (IND), and insulin (INS), each of these playing its role in adipogenesis (Figure 4). Firstly, the combination of DEX and IBMX activates PPARγ expression, which induces adipogenic differentiation [78]. DEX is a steroidal anti-inflammatory molecule, while IBMX is a nonselective phosphodiesterase inhibitor that acts by elevating the intracellular levels of cyclic AMP (cAMP) and protein kinase A (PKA), both signaling pathways required for transcriptional activation of PPARγ [77]. IND, a nonsteroidal anti-inflammatory exogenous ligand that enhances PPARγ expression, also promotes PPARγ activation, leading to the regulation of transcription factors from the CEBP family, such as CEBPA. Once activated, CEBPA forms a positive feedback loop with PPARγ, therefore upregulating each other’s expression levels [78]. Although not strictly necessary for adipogenesis initiation, INS plays a significant role in the late stages of differentiation by acquiring the metabolically active mature adipocyte phenotype. For example, a variety of paracrine and endocrine activities of molecules such as leptin and adiponectin, which are classically expressed in adipocytes, are insulin-regulated and, therefore, dependent on its presence for full maturation [79]. 

Notably, most adipogenic differentiation protocols currently available have been developed and standardized in two-dimensional (2D) cell models, such as 3T3-L1 cells and mesenchymal stem cells (reviewed by [80]), which will be discussed in the following sections.

### 4.3. D Culture as a Key Model for Adipogenesis Studies

2D adipogenic differentiation is widely used because it is a relatively easy and cost-effective technique [81,82]. Although this model does not reflect the physiological situation in their tissues of origin, it is still a widely used practice, especially in the early stages of research projects.

The murine cell line 3T3-L1 remains the most frequently used cell model for adipogenesis assessment and obesity-related studies. These cells exhibit typical fibroblast morphology that can be converted into adipocyte phenotype by elevating the intracellular levels of cyclic AMP (cAMP) with a cocktail of adipogenic inducers, such as DEX (1 µg/mL), INS (0.25 µM), and IBMX (0.5 mM) after 14 days in culture [11]. In association with the adipogenic medium, other pro-differentiation agents may enhance adipogenesis in 3T3-L1 cells. The addition of rosiglitazone (2 µM) to the differentiation medium already containing DEX, INS and IBMX was reported to generate complete mature adipocytes within 10–12 days of treatment, which could still be observed after ten passages [83]. Similarly, the differentiation efficiency of frozen-thawed 3T3-L1 cells was improved after prolonged treatment with rosiglitazone (2 µM) in the presence or absence of IBMX. However, this improvement also depended on the concentration of other adipogenic inducers applied in the medium [84]. 

Given the ease of maintenance in comparison to freshly isolated adipocytes, the capacity to survive multiple passages, and the high homogeneity of the cell population [11], 3T3-L1 cells have been used for the evaluation of substances that regulate adipogenic differentiation [85], as well as in the discovery of anti-obesity agents [86] and endocrine disruptors that act upon this differentiation process [87]. For example, aiming to discover AOM based on naturally sourced oleanolic acid derivatives, adipogenesis in vitro assays conducted with 3T3-L1 cells resulted in the finding of a novel compound, HA-20, with potent inhibitory activity on adipogenesis. It was also demonstrated that HA-20 markedly suppressed the adipogenesis in 3T3-L1 at the early stage without cytotoxicity. Thus, it may serve as a leading compound for further developing anti-obesity agents [88]. Some other compounds have also been evaluated using 3T3-L1 cells, such as the antidiabetic agent lansoprazole, which enhanced adipocyte differentiation at low concentrations via PPARγ and CEBPA activation and lipogenic protein expression of ACC1 and FASN [89].

Increasing the adipocyte number subsequently increased the basal and insulin-stimulated glucose uptake and expression of GLUT4 mRNA. Interestingly, on the other hand, high concentrations of lansoprazole strongly inhibited differentiation and expression of PPARγ and CEBPA and maintained the expression of the preadipocytes markers, β-catenin, and PREF-1 [89]. It is also possible to seek new ligands with high throughput screening, such as the antifungal drug fenticonazole nitrate, which has been identified as a new potent PPARγ-modulating ligand to significantly demonstrate antidiabetic and anti-nonalcoholic fatty liver disease, using, among others, adipocyte differentiation assays with 3T3-L1 cells and molecular docking validation [90]. However, the results obtained with animal models and non-human cell lines may not be adequately translated to humans due to interspecies differences [91], a significant drawback of 3T3-L1 utilization for obesity studies.

In this context, much has been discussed about the applicability of MSCs in adipogenesis and obesity modeling. Besides their multipotency and capacity to expand in culture, they can be isolated from a variety of human sources, including adipose tissue, therefore being able to display donor and depot-specific characteristics (reviewed by [10]). The in vitro adipogenic differentiation of MSCs is usually achieved after 14 days of treatment with a culture medium containing IBMX (1 mM), INS (2 µg/mL), DEX (2 µM), and IND (400 µM), which should be renewed every 3 to 4 days [92]. The quantification of lipid droplet formation in MSCs has also improved with advancements in high-content imaging analysis and the possibility to measure parameters such as cell differentiation area and percentage of adipogenesis [93].

MSCs have been used as a model for evaluating various compounds during adipogenesis. For example, Abud and colleagues have established an inhibition adipogenesis assay for the toxicological prediction of possibly adipogenic disruptors chemicals [92,93]. The authors observed a high-fidelity prediction rate for substances considered toxic in the GHS scale using adipogenesis as an endpoint. Similarly, Ribeiro and colleagues observed impairment in the adipogenic differentiation of MSCs after treatment with bismuth nanoparticles that were not previously detected in undifferentiated MSCs, confirming the sensibility of this process for toxicity evaluation [94]. Moreover, MSCs have served as a tool to investigate the impacts of endocrine disruptors on adipogenic differentiation and fat development [95] and the anti-adipogenic [20,96,97] or pro-adipogenic [98] effects of several compounds. 

Thus, 2D models have been demonstrated to be useful for AOM research and the study of molecular pathways regulating adipogenesis in murine and human cells, as reviewed [11,80]. Both models can contribute to the study of obesity, including the endocrine disruptor effect described for many compounds (reviewed by [99]). As there is always room for improvement, recently developed adipogenesis models can take anti-obesity research even further, such as the case of 3D organoids derived from human adipose stem cells for adipose tissue physiology studies [100]. 

#### 4.3.1. Morphological Evaluation of In Vitro Adipogenesis

During the process of adipogenesis, progenitor cells undergo several morphological changes due to the restructuring of the cytoskeleton and changes in organelles. These changes lead to an adequate cell shape and structure to accommodate the lipid droplets. These changes include actin fiber rearrangement and mitochondrial biogenesis. The transition from the fibroblastoid to the spherical shape, characteristic of mature adipocytes, is related to decreased focal adhesions, stimulating adipogenic differentiation. It is possible to verify these alterations through immunofluorescence analyses. It is also possible to distinguish the degree of maturity of the adipocytes through the ultrastructural analysis of lipid vesicle size and quantity [101,102]. The most prominent feature of adipogenesis is lipid vesicle formation, and a variety of dyes and fluorescent probes can be used to stain lipid droplets. 

Oil red—O (C_26_H_24_N_4_O) is a lipid-soluble diazolic dye that detects neutral lipids. It has maximum absorption at 518 nm, is selective for neutral lipids and cholesteryl esters, and does not stain biological membranes. The principle of the method is that Oil red—O is hydrophobic and moves from its solvent to associate with lipids [103,104].

Nile red was synthesized from the Nile blue oxidation and is a hydrophobic and metachromatic dye with low solubility. This fluorochrome is fluorescent in an aqueous medium and does not dissolve lipids. Color emission varies from deep red to intense yellow in hydrophobic environments. Depending on the excitation and emission wavelength, this dyeing is used for different hydrophobic molecules. This probe has the advantage that it can be used to distinguish neutral lipids from phospholipids or other amphipathic lipids [105,106,107]. 

Bodipy lipid probe (boron-dipyrromethene, excitation 450–490 nm and emission 515–530 nm) is a marker for fat accumulation in live cells that is insensitive to the environment polarity, in contrast to Nilo Red. Numerous studies have established the specificity of Bodipy 505/515 for lipid droplets [107,108,109,110]. 

Currently, there is the possibility of analyzing, in an automated way, the adipocytes in culture after cytochemistry and immunofluorescent staining, which facilitates and complements the analyses previously discussed [93,111].

#### 4.3.2. D Systems Mimetic the Organ Architecture

3D cultures have been explored as models for obesity research since they provide conditions that recapitulate human physiological systems, allowing cell–cell interaction and cell-matrix organotypic relief [112]. Different techniques can be used to create three-dimensional structures that seek to resemble in vivo systems, such as organoids, scaffolds as constituent supports of hydrogel, polymeric material, and fiberglass, as reviewed [113].

It has been pointed out that, in 3D cultures, 3T3-L1 strains and differentiated MSCs acquire more mature adipocyte phenotypes compared to 2D cultures. Thus, they constitute a promising strategy to cultivate adipocytes for an extended period since cellular displacement is observed as the adipocytes undergo maturation [114].

Graham and collaborators developed a high-throughput model using spheroids composed of 3T3-L1 white adipose tissue cells encapsulated in a 3D matrix, proving to be a better drug-responsive model than 2D cultures. The spheroid model demonstrated a significant upregulation of certain specific adipogenic genes such as PPARγ, FABP4, ADIPOQ, and FASN compared to a 2D model after 14 days of differentiation [115]. An additional 3D culture model was developed recently to create an in vitro system for obese adipocyte dysfunction research. The hMSCs were embedded in hydrogel and differentiated into adipocytes within a thin cellulose scaffold that allowed 3D tissue modeling. The cells were further treated with oleic or palmitic acid to simulate a caloric overload. It was demonstrated that the adipocytes presented with larger lipid droplets and metabolic profiles compatible with hypertrophy, such as increased basal lipolysis and insulin resistance. Thus it proves to be a valuable tool in the studies of adipocyte dysfunctions because of obesity [116]. 

Shen and collaborators developed a platform containing adipocytes in 3D culture with chemically predefined conditions. The authors verified that after the differentiation induction, the spheroids presented with molecular and cellular phenotypes that resembled freshly isolated mature adipocytes. The cells remained phenotypically stable for up to one and a half months [112]. Recently, in the search for optimization in the drug screening models to be applied in the context of obesity-related metabolic syndrome, 3T3-L1 cells were induced to undergo adipogenesis on 3D alginate hydrogel beads. Adipocytes were co-cultured with macrophages to simulate an inflammatory environment and compared with 2D cultures. After the induction of cells into adipocytes, they perceived the characteristics of inflamed adipose tissue with an accumulation of lipid droplets in the cells. The co-cultured cells presented the characteristics of insulin resistance, which did not occur with 2D cultures. By adapting the 3D system into a microwell format, the authors demonstrated that it could be used as a sensitive high-throughput assay for drug screening. This study indicates that the assigned technique could be used to screen other similar metabolic diseases [117]. 

Aiming to compare hMSC from 2D and 3D cultures, spheroid constructs were developed using magnetizable nanoparticles during adipogenic differentiation. Wolff and colleagues investigated the release of adipokines from cells in the differentiation process for up to 28 days of the assay [118]. They verified its influence on cellular development in different conditions, the occurrence of adipogenic differentiation, and the similar liberation of adipokines in both conditions. The release of adiponectin and leptin was considerably higher in 3D spheroids, increasing the progression of the period of adipogenic stimulation. On the other hand, a decrease in the release of pro-inflammatory factors such as IL-6 and IL-8 was observed. Based on this study, one can consider that the three-dimensional model of culture is a reliable tool for studying MSCs, since uniform, symmetrical and long-lasting spheroids were obtained. Even with current studies demonstrating the potential of 3D cultures, the correlation with in vivo models still needs to be improved, with cells presenting considerable multilocularity, low adipokines intensity, and limited lipids accumulation [112]. Thus, exploring this field with great potential to contribute to such studies of interest is still necessary.

### 4.4. Bioprinting as a New Tool for 3D Systems

Another promising and innovative strategy that could help to improve the construction of 3D tissues is the 3D bioprinting technology. Briefly, bioprinting consists of fabricating a construct using a bioink (cells embedded in a hydrogel or other support components) to build an organ or tissue of interest with a specific structure and format [119]. Some studies used this approach to print adipose tissue, but usually, the focus is to generate adipose mimetic tissues for soft/adipose tissue regeneration. On the other hand, some works seek to understand whether MSC could be bioprinted in different materials with different stiffness and maintain their potential for differentiation [120,121,122,123]. However, it is possible to learn from these approaches and apply them in other contexts, such as drug research.

One interesting strategy to generate adipose microtissues is to combine the formation of spheroids from MSCs (3D cultures) with a printable hydrogel or matrix. These studies used spheroids with only MSCs [124] or associated them with other cell types as endothelial cells to allow vascularization of the tissue [125] or with breast cancer cells to create an interface to study the interaction between these cells and adipose cells [126]. In general, they all showed that adipose-derived stem/stromal cells (ADSCs) could differentiate into adipocytes in the 3D structure with equal or better effects than 2D cultures or spheroids only. However, considering the necessity of developing larger structures or cell cultures that better mimic the native tissue, bioprinting is a good option.

Considering the necessity of better mimicking adipose tissue, Pati and coworkers, in 2015, used a bioink composed of a decellularized extracellular matrix (dECM) of adipose tissue with ADSCs and printed a structure that could lead to adipogenic differentiation better than the unprinted combination. Interestingly the dECM alone (without adipogenic induction medium) induced the expression of adipogenic markers in ADSCs. In addition, the 3D construct was implanted subcutaneously in mice showing integration with the tissue [127].

Other approaches are still used in the search to generate adipose tissue in vitro. For example, using laser-assisted bioprinting technology, 3D grafts were generated with ADSCs or with already differentiated ADSCs [128]. In addition, using a cryogel (that allows adipogenic differentiation) as support, Qi et al. printed using two bioinks, alternating the layers of the structure, one with pre-differentiated ADSCs encapsulated in Me-HA/Me-Gel, and another layer with endothelial cells (HUVEC) encapsulated within a fibrin-based gel. These allow the generation of vascularized structures with adipocytes [129].

More recently, a study from Zhang and collaborators aimed to develop a 3D in vitro model to study, among others, AOM. The group used a co-axial co-culture bioprinting methodology, combining neural cells and ADSCs to create the tissue. The cells were shown to differentiate into adipocytes. Still, in the presence of neural cells, the lipid droplets were smaller, with higher levels of FABP4 and adiponectin, indicating that neural cells could improve fat metabolism. Thus, this could be an exciting approach to future anti-drug research [130].

Thus, we observed that although bioprinting is a recent technology, it has already been applied to adipose tissue engineering, generating 3D structures capable of adipogenic differentiation that could be used for AOM research. However, further studies are needed to better understand the cell behavior in 3D bioprinted tissues and to confirm the best strategy for printing and generating tissues that are more similar to adipose tissue physiology (e.g., vascularization, innervation).

### 4.5. Microfluidics

The breakthrough technology of microphysiological systems (MPS) has been changing preclinical research by allowing the successful development of biologically relevant human models [131]. The possibility of cultivating human cells in a very controlled microenvironment, mimicking physiological functions of tissue and organs, permits the performance of more physiologically relevant, predictive, and ethical methods to be used during drug discovery and development [132]. MPS, also frequently known as organ-on-a-chip, are usually defined as miniaturized cell culture platforms containing human or animal cells and tissues in a way to reproduce the physiological functions of one or more organs. In practice, these are devices fabricated from different materials containing living cells organized in 2D or 3D architectures and maintained in finely controlled conditions. Often, these devices have a microfluidic system containing perfused hollow microchannels combined with culturing chambers allowing the reconstitution of tissue interfaces and organs microenvironments and functions [133]. The MPS field is growing fast, and the area has focused mainly on cardiac, hepatic, neuronal, and oncology models, which are of great interest for drug development purposes. However, some researchers have suggested the importance of developing adipose tissue MPS either in association with other organs to investigate the influence of adipose tissue in the pharmacotoxicological profile of drugs or separately in a system to study the pathophysiology of obesity and fat-related diseases [134].

As reviewed by McCarthy and colleagues, a diversity of fat-on-a-chip models has been proposed to reproduce metabolically active adipose depots to be applied in the search for novel small molecules or biologics as potential treatments for fat-related diseases. The tissue complexity must be simulated to achieve a functional, healthy, or diseased fat microenvironment. Such robustness is sought by applying primary and/or cell lines cultivated into various bioscaffolds and sometimes under dynamic cultivation using bioreactors. The greater the diversity of components and complexity, the greater the probability of obtaining a genuinely predictive system [135].

One common approach is combining immune cells with adipocytes to mimic the inflammatory character of obesity-associated diseases such as type II diabetes mellitus. On-chip dynamic co-cultures of preadipocytes with peripheral blood mononuclear cells (PBMC) show a diabetes-like pattern of cytokine profile and insulin resistivity. The model’s functionality was proven by challenging the system with the antidiabetic drug metformin and the nutraceutical omega-3, which reduced inflammation and improved insulin sensitivity in a similar physiological way [136]. Other models were proposed to study insulin resistance using preadipocytes and macrophages cultivated into microfluidic devices. Studies have demonstrated an adipocyte-immune co-culture in a planar compartmentalized microfluidic system with porous sidewalls. Each type of cell was cultivated in separate interconnected compartments, and different designs of fluidic compartments were tested. First, preadipocytes were differentiated into adipocytes under dynamic medium flow conditions, being the medium infused by adjacent channels. Differentiated cells were then exposed to a physically separated co-culture with immune cells, and the induction of an inflammatory environment with lysophosphatidic acid challenged the system. All cell-cell crosstalk happened by continuous exchange of bio/chemical signals which was allowed by the porosity of the sidewalls. The authors show the efficiency of the system as a diabetes mellitus type-2 model by showing cytokines release and glucose uptake and suggest that the system could be applied to model other parenchymal-immune cell co-cultures [137,138]. Such systems prove the feasibility of physiologically relevant models as valuable tools to study type II diabetes mellitus mechanisms and diabetic drug screening.

Besides adipocyte-based models, some studies have proposed using (ADSCs) as a substrate for adipose MPS. For instance, O’Donnell and colleagues demonstrated that ADSCs cultivated into methacrylated gelatin constructs in a 3D-printed bioreactor maintained their capacity for adipogenic differentiation under continuous perfusion. The authors suggested that the platform could be used as an efficient microphysiological system to study the adipose tissue’s influence on osteoarthritis pathophysiology [139]. Multicomplex microphysiological systems allow the reproduction of physiological features, such as tissue architecture and biochemical signaling, which are essential for studying complex physiopathological conditions like adipose tissue-related diseases [135]. However, microfluidic chip technology can also be applied in simpler contexts where the possibility of automated and precise control of the cellular microenvironment improves the robustness of cell models. An excellent example of such an approach was the development of a microfluidic large-scale integration chip platform containing 128 separated microchambers for long-term culturing and differentiation of hMSCs. According to the authors, differentiation into the chip resulted in more homogenous outcomes when compared to adipogenesis in 96-well plate culturing. For assessing lipid accumulation, a high-content image system was used. In addition, the authors also proposed an in situ protein analytics method through a multiplex in situ proximity ligation assay to overcome the difficulty of the limited analytical capability of common microfluidic platforms. This chip platform allowed the precise investigation of the molecular role of the mammalian target of rapamycin (mTOR) during adipogenesis [140].

Although 2D cell culture models of adipogenic differentiation are useful and largely applied in understanding the development, function, and modulation of adipose tissue, studies have suggested that in 3D models, the complexity of adipose tissue is better mimicked, and in vitro adipogenesis is optimized. Yang and colleagues demonstrated that on-chip differentiation of stem cells into a 3D adipose microtissue was possible. For that, hMSCs were loaded into the chip chambers in a solution of fibrinogen and thrombin. After polymerization, 3D constructs were exposed to an adipogenic differentiation medium through adjacent channels, and after differentiation, the adipocyte’s function was challenged under interstitial flow [141]. The same system was used to perform a 3D co-culture of hMSCs and endothelial cells in an attempt to achieve a vascularized human adipose tissue model. The co-culture medium’s composition needed to be adapted to keep the cells’ angiogenic and adipogenic potential. Moreover, pre-induction of the hMSCs in the 2D environment before loading into the chip increased the system’s efficiency. With the modifications, it was possible to assemble a functional vascularized human adipose tissue model ready to be used in adipose tissue-related studies [142]. Fat-on-chip models, simpler or more complex, are undoubtedly becoming an accessible and valuable tool to the research community, either to study the physiopathology of adipose tissue alone or in combination with other organoids to be applied in drug discovery, mainly of fat-related diseases [135]. Although still not broadly exploited, hMSCs adipogenic differentiation on-chip is a powerful method to investigate adipose tissue-related biology and diseases precisely.

## 5. Challenges and Limitations

Despite advances in clinical studies for obesity and associated metabolic diseases and the establishment of different therapeutic targets, this area of research still suffers from several limitations. Although large, the repertoire of genes, proteins, molecules, or even metabolic pathways presents a lack of knowledge about their interactions and specific functions during adipogenesis and processes associated with obesity. Moreover, the complex redundancy of these players makes it difficult to establish reliable markers for obesity and related metabolic diseases [5,143]. Still, although studies point to a possible link between obesity and different markers, it is not clear whether these markers, once altered, lead to obesity or whether the onset of obesity is the trigger for the differential expression of these markers. Further mechanistic studies are needed to understand these causal relationships between obesity and such markers [144].

Moreover, there are many endpoints to evaluate regarding obesity studies, considering the complexity of the disease. In this scenario, the systemic and random molecular docking simulations could lead to an exponential number of conformations that need to be analyzed, which may not be practical. Some algorithms refine these data but are generally expensive and time-consuming. In addition, the computation resource for the in silico docking approaches using DNA is still limited and challenging [145,146,147].

In addition, as in the development of any new drug, the stages of clinical trials can be difficult and require careful study, mainly because in vitro and in silico methods have several limitations concerning the organism of in vivo models and humans. Despite progress, common problems seen in the clinical phase are to be expected, such as inadequate bioavailability, impractical pharmacokinetic or pharmacodynamic characteristics, and adverse effects on patients [91,143].

## 6. Future Directions

To date, basic and clinical research has greatly advanced our understanding of obesity as a chronic disease, with multiple organs involved and under the influence of genetic and environmental factors. However, there is still an unmet need for effective anti-obesity therapies. Drugs that specifically modulate proteins and transcription factors expressed explicitly in adipocytes constitute a promising approach to treating obesity. The individual responses to available treatment may represent an opportunity for studies in precision medicine applied to obesity research [148], also investigating the effects of xeno-free systems [149] and the development of more physiological systems to address these individual responses. Besides, the differentiation, hypertrophy, and physiological balance of white, beige, and brown adipose tissue may shed light on the obesity process and the involvement with neural [150] and endocrine system regulators.

As PPARγ and CEBPA family members are the central modulators of adipogenesis, they are widely studied targets for the in vitro and in vivo studies of anti-obesogenic medicine. Therefore, insights into various signaling pathways, energy-sensing proteins (e.g., AMPK), genes, and their transcription factors that directly interact with PPARγ and CEBP family members are needed to combat abnormal adipose tissue development during the obesity pandemic.

Additionally, AOM discovery may benefit from artificial intelligence approaches [151], and other regulatory systems of gene expression, such as microRNAs [152,153,154] or long non-coding RNAs [155], may be investigated in the future.

## 7. Conclusions

Overweight and obesity represent major health threat that has grown to epidemic proportions as their rates continue to rise in adults and children in all countries

A better understanding of the adipogenesis process may lead to new strategies to improve metabolic health in obesity, identifying potential therapeutic targets. Identifying the up and downregulated genes during adipogenesis, lipogenesis, and lipolysis are relevant tools for obesity research in vitro. Studies of this nature can be, and have been, used as endpoints to investigate the potential of AOM and to assess their effects during and post-adipogenesis.

The use of in vitro models of adipogenesis has been increasing, mainly in two-dimensional (2D) cellular models. However, results obtained with non-human cell lines may not be adequately translated to humans due to interspecies differences. For this reason, there is a growing interest in the applicability of MSCs in adipogenesis and obesity modeling. Other promising and successful technologies in this context are 3D cultures and tissues and microphysiological systems (Figure 5).

In silico methodologies have also been widely applied in researching several diseases and biological processes, including adipogenesis and metabolic syndromes. For example, gene enrichment analysis, the search for interactions between signaling molecules and biomarkers, and molecular docking allow the identification of pathways and sites of interactions between proteins and small molecules and drug relocation. All these data obtained may provide new information and serve as a basis for future studies.

In this article, we seek to shed light on the main alternative models for the study of adipogenesis, aiming to contribute to preclinical research in obesity. As adipogenesis plays a key role in determining the differentiation of adipocytes, it is a possible therapeutic approach for obesity. Identifying transcriptional cascades, signaling pathways, and molecular mechanisms during this process may serve as potential therapeutic targets. The topics addressed in this review may help new studies search for data on the safety and metabolism of possible anti-obesity treatments.

## Figures and Tables

**Figure 1 life-13-00108-f001:**
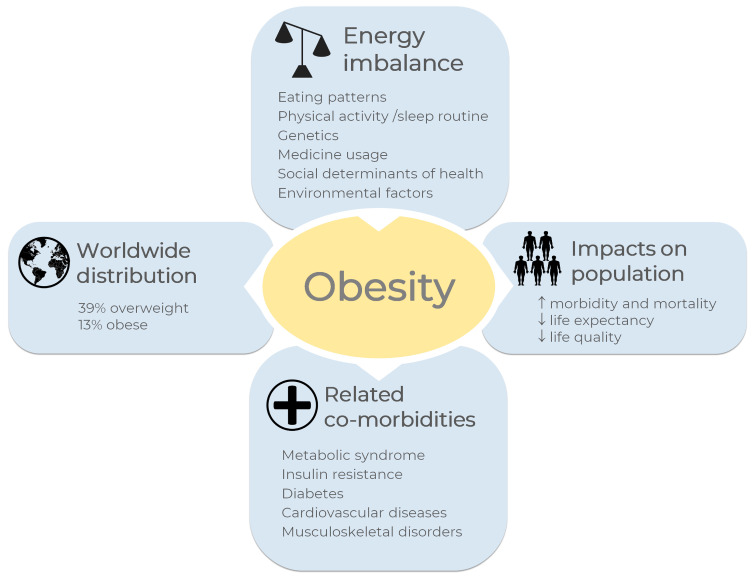
Overview of obesity in the world. A general definition for obesity and overweight is enhanced fat storage due to imbalanced energy between calories consumed and expended. Eating patterns, physical activity levels, sleep routines, and genetics, among others, are the main causes of energy imbalance. According to the World Health Organization, one billion and nine hundred million adults worldwide were overweight in 2016; of these, 650 million were obese [1], corresponding to 39% and 13% of the world population, respectively. Some diseases are correlated with obesity, such as metabolic syndrome, insulin resistance, diabetes, cardiovascular diseases, musculoskeletal disorders, and some types of cancer. Obesity increases morbidity and mortality rates and reduces life quality and life expectancy. The images were obtained from Servier Medical Art 266 (http://smart.servier.com/, accessed on 21 December 2022), licensed under a Creative 267 Commons Attribution 3.0 Unported License.

**Figure 2 life-13-00108-f002:**
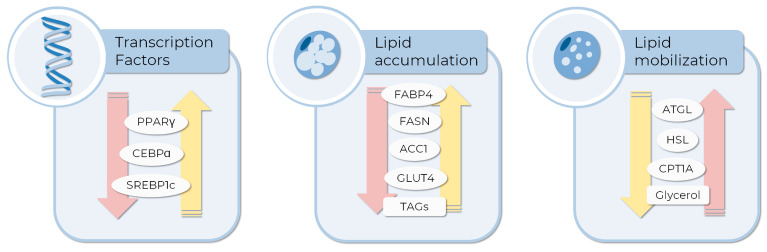
Molecular and metabolic endpoints to evaluate the anti-obesity compound or pro-adipogenic differentiation activity of drugs in MSCs in vitro testing. The expression of some transcription factors inducing adipogenesis, such as PPARγ, CEBPA, and SREBP1c, are currently evaluated as differentiation markers. Moreover, the expression or activity of enzymes related to glucose and lipid metabolism can be employed to measure the extent of differentiation or lipid consumption induced by tested compounds. Yellow and pink arrows represent the expected expression pattern of each endpoint to pro-adipogenic and AOM, respectively. The images were obtained and adapted from Servier Medical Art (http://smart.servier.com/, accessed on 11 December 2022), licensed under a Creative Commons Attribution 3.0 Unported License.

**Figure 3 life-13-00108-f003:**
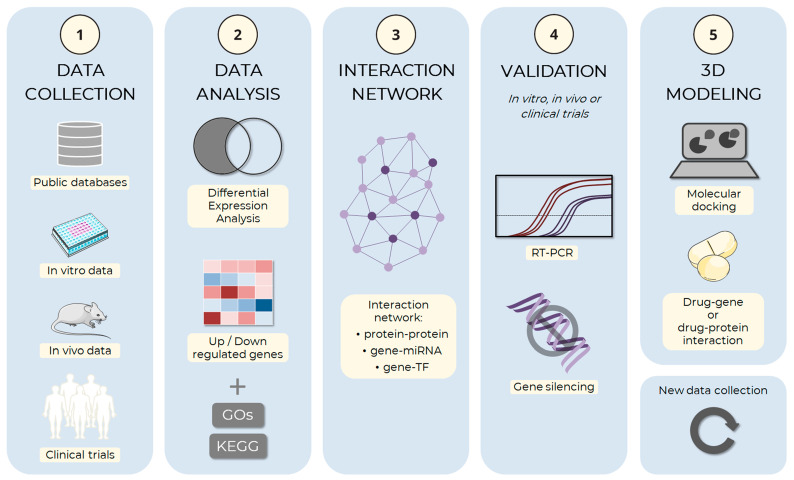
Schema of the adipogenesis in silico analysis pipeline. The computational simulation process of adipocyte development can be divided and implemented in an analysis pipeline. (**1**) The first step of the process is the collection of data on adipogenesis, including information about genes, proteins, and molecular mechanisms involved in the process. Data can be obtained from public databases, such as the Gene Expression Omnibus (GEO), ArrayExpress, and Sequence Read Archive (SRA), or from experimental approaches, such as in vitro and in vivo assays, or clinical trials, among others. (**2**) The collected data are analyzed to identify the genes and proteins involved in adipogenesis, the molecular mechanisms involved in the process and the regulatory factors, and which ones are up and down-regulated. During the gene expression analysis step, Gene Ontology (GO) and the Kyoto Encyclopedia of Genes and Genomes (KEGG) can be used to identify genes being expressed under different conditions and understand how these genes may be involved in adipogenesis processes. (**3**) After selecting the main factors, the hub genes are identified through the analysis of interactions between the factors: protein–protein interaction, target gene–miRNA interaction, or target gene–transcription factor (TF) interaction, among others. (**4**) Those with the highest degree of connections should preferably be selected for in vitro or in vivo experimental validation, which may use RT-PCR, gene silencing, or other techniques. (**5**) Finally, the main selected factors can be used from basic research to develop new drugs through 3D modeling methods, both the target (gene or protein) and small therapeutic molecules. The images were obtained and adapted from Servier Medical Art (http://smart.servier.com/, accessed on 22 December 2022), licensed under a Creative Commons Attribution 3.0 Unported License.

**Figure 4 life-13-00108-f004:**
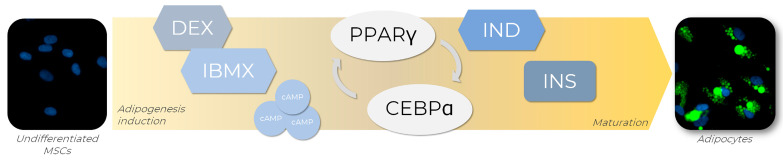
Adipogenic differentiation of MSCs is promoted by adipogenic inducers. To achieve adipogenesis in vitro, MSCs receive an adipogenic cocktail containing dexamethasone (DEX) and isobutyl-1-methylxanthine (IBMX), indomethacin (IND), and insulin (INS), which are responsible for the activation of key transcription factors involved in the adipogenesis regulatory cascade, such as PPARγ and CEBPA. The transcriptional activation of PPARγ is triggered by DEX and IBMX through the elevation of intracellular levels of cyclic AMP (cAMP) and positive regulation of the proteinase kinase A (PKA) signaling pathway. Concomitantly, PPARγ expression is enhanced by IND exposure, leading to CEBPA regulation. Once activated, CEBPA participates in a positive feedback loop with PPARγ, regulating each other’s expression levels. In the late stages of differentiation, INS promotes the acquirement of metabolic phenotype characteristics of mature adipocytes. AD-MSCs (Lonza^®^, Walkersville, MD, USA; catalog number PT-5006) were cultivated for 14 days in the absence (undifferentiated MSCs) or presence (adipocytes) of adipogenic inducers and differentiation medium. Lipid droplet formation (green—Nile Red) and nuclei (blue—DAPI) images were obtained with a high-content imaging system (Operetta CLS^®^, PerkinElmer^®^, Waltham, MA, USA).

**Figure 5 life-13-00108-f005:**
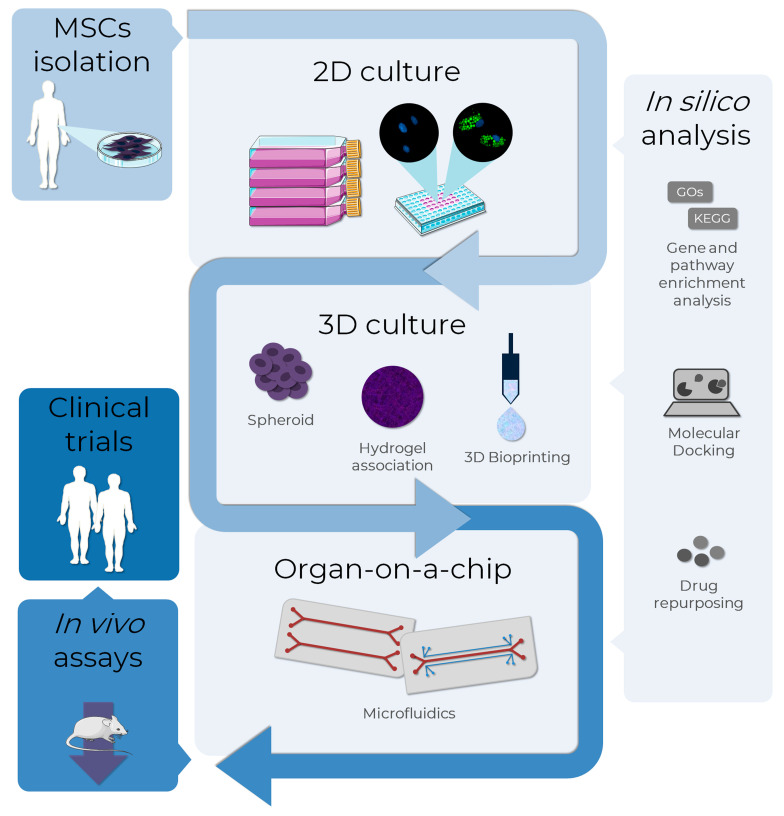
MSCs as an alternative method to test adipogenesis-related drugs in obesity research. The MSCs, isolated from several sources as adipose tissues, can be used as in vitro models to evaluate the pro-adipogenic or anti-obesity potential in drug discovery. Bidimensional cultures represent the most employed in vitro method, although it is the least representative in terms of physiological conditions. To overcome this, 3D culture methods such as spheroid formation, and the association of cells to biomaterials, such as hydrogels and bio-printing, are approaches currently used to improve cell-cell and cell-extracellular matrix interactions, resembling the human cells environment. A more complex and biologically-relevant model to mimic the physiological functions of tissues or organs is the organ-on-a-chip technology, where cells are cultivated in 2D or 3D in microdevices under a controlled microenvironment. In silico analysis assists all the in vitro stages with bioinformatic tools: databases and enrichment analysis software allow the analysis of high throughput data, while molecular docking and drug repurposing help in the discovery of drugs to specific targets. The integrated use of these alternative methods allows the translation from in vitro to in vivo tests using a reduced number of animals in the search for new AOM, also providing more information that can support clinical trials. AD-MSCs (Lonza^®^, Walkersville, MD, USA; catalog number PT-5006) were cultivated for 14 days in the absence (undifferentiated MSCs) or presence (adipocytes) of adipogenic inducers and differentiation medium. Lipid droplet formation (green—Nile Red) and nuclei (blue—DAPI) images were obtained with a high-content imaging system (Operetta CLS^®^, PerkinElmer^®^, Waltham, MA, USA). The images were obtained and adapted for the schematic representation from Servier Medical Art (http://smart.servier.com/, accessed on 11 December 2022), licensed under a Creative Commons Attribution 3.0 Unported License.

## Data Availability

Not applicable.

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
