# Peer review of "Alternative Methods as Tools for Obesity Research: In Vitro and In Silico Approaches"

_life, 2022, doi:10.3390/life13010108_

Round 1

Reviewer 1 Report

Comments to Authors 

            This study showed that: a) to provide the highlights of the adipogenesis study based on in vitro differentiation of human mesenchymal stem cells (hMSCs), listing in silico methods, such as molecular docking for identification of molecular targets, and in vitro approaches, from 2D, more straightforward and applied for screening large libraries of substances, to more representative physiological models such as 3D and bioprinting models; b) to describe the development of physiological models based on microfluidic systems applied to investigate adipogenesis in vitro.

          Authors are kindly requested to emphasize the current concepts about these issues in the context of recent knowledge and the available literature. This article should be quoted in the References list.

References

1.      Influence of Obesity in the miRNome: miR-4454, a Key Regulator of Insulin Response Via Splicing Modulation in Prostate. J Clin Endocrinol Metab. 2021;106(2):e469-e484. doi:10.1210/clinem/dgaa580.

2.      MicroRNAs as Mediators of Adipose Thermogenesis and Potential Therapeutic Targets for Obesity. Biology (Basel). 2022;11(11):1657. Published 2022 Nov 13. doi:10.3390/biology11111657.

Author Response

Response to Reviewer 1 Comments

Point 1: This study showed that: a) to provide the highlights of the adipogenesis study based on in vitro differentiation of human mesenchymal stem cells (hMSCs), listing in silico methods, such as molecular docking for identification of molecular targets, and in vitro approaches, from 2D, more straightforward and applied for screening large libraries of substances, to more representative physiological models such as 3D and bioprinting models; b) to describe the development of physiological models based on microfluidic systems applied to investigate adipogenesis in vitro.

          Authors are kindly requested to emphasize the current concepts about these issues in the context of recent knowledge and the available literature. This article should be quoted in the References list.

References

  1. Influence of Obesity in the miRNome: miR-4454, a Key Regulator of Insulin Response Via Splicing Modulation in Prostate. J Clin Endocrinol Metab. 2021;106(2):e469-e484. doi:10.1210/clinem/dgaa580.
  2. MicroRNAs as Mediators of Adipose Thermogenesis and Potential Therapeutic Targets for Obesity. Biology (Basel). 2022;11(11):1657. Published 2022 Nov 13. doi:10.3390/biology11111657.

 Response 1: We thank the reviewer for the comments that helped to improve the manuscript. We agree with the reviewer since microRNAs are mediators of obesity and their role in adipogenesis is very relevant, as it has been extensively studied and reviewed in the literature. We have read the suggested references and addressed this topic in the section Future directions.

Additional Comments:

We also have made some updates to the manuscript regarding the spellings of the corresponding author

From Alessandra Melo De Aguiar to Alessandra Melo de Aguiar

And we also included some more information in the figure caption, below (highlighted in red):

Figure 5: MSCs as an alternative method to test adipogenesis-related drugs in obesity research. The MSCs, isolated from several sources as adipose tissues, can be used as in vitro models to evaluate the pro-adipogenic or anti-obesity potential in drug discovery. Bidimensional culture represents the most employed in vitro method, although the least representative in terms of physiological conditions. To overcome this, 3D culture methods such as spheroid formation, association of cells to biomaterials as hydrogels and bio-printing are approaches currently used to improve cell-cell and cell-extracellular matrix interactions, resembling the human cells environment. A more complex and biologically-relevant model to mimic physiological functions of tissues or organs is the organ-on-a-chip technology, where cells are cultivated in 2D or 3D in microdevices, under a controlled microenvironment. In silico analysis assists all the in vitro stages with bioinformatic tools: databases and enrichment analysis software programs allow the analysis of high throughput data, while molecular docking and drug repurposing help in the discovery of drugs to specific targets. The integrated use of these alternative methods allows the translation from in vitro to in vivo tests using a reduced number of animals in the search for new AOM, also providing more information that can support clinical trials. AD-MSCs (Lonza®, Walkersville, USA; catalog number PT-5006) were cultivated for 14 days in the absence (undifferentiated MSCs) or presence (adipocytes) of adipogenic inducers and differentiation medium. Lipid droplet formation (green – Nile Red) and nuclei (blue – DAPI) images were obtained with a high-content imaging system (Operetta CLS®, PerkinElmer®, Waltham, MA, USA). For the schematic representation, the images were obtained and adapted from Servier Medical Art (http://smart.servier.com/, Accessed on 11/12/2022), licensed under a Creative Commons Attribution 3.0 Unported License.

We have added the term “and adapted” to all the figures made with images from Servier Medical Art, once we have made some adaptations to the original images.

Reviewer 2 Report

The authors aimed to identify the main alternative models, contributing to the direction of preclinical research in obesity. They concluded that in silico and in vitro techniques may provide a clear picture of alternative methods based on adipogenesis as a tool for obesity research.

This is an excellent, well-structured, and well-written review with useful figures and complete reference list.

Comments:

1.     The list of keywords seems to be incomplete.

2.     PPARG should be corrected to PPARγ.

3.     Some further figures may improve the value of the review.

4.     English needs minor editing.

Author Response

Response to Reviewer 2 Comments

Point 1: The authors aimed to identify the main alternative models, contributing to the direction of preclinical research in obesity. They concluded that in silico and in vitro techniques may provide a clear picture of alternative methods based on adipogenesis as a tool for obesity research.

This is an excellent, well-structured, and well-written review with useful figures and complete reference list.

Response 1: We thank the reviewer for the positive comments and kind words.

Point 2: The list of keywords seems to be incomplete.

Response 1: We have included some more keywords.

Keywords: obesity; adipogenesis; mesenchymal stem cells; human adipose derived stem cells; in vitro alternative methods; in silico alternative methods, 3D culture, microfluidics

Point 3: PPARG should be corrected to PPARγ.

Response 1: We have changed PPARG to PPARg. We have previously used the term PPARG as it is the official symbol of the gene (PPARG peroxisome proliferator activated receptor gamma [Homo sapiens (human)] - Gene - NCBI (nih.gov)

Point 4: Some further figures may improve the value of the review.

Response 1: We have included two additional figures to the manuscript

Figure 1: Overview of obesity in the world. A general definition for obesity and overweight is enhanced fat storage due to imbalanced energy between calories consumed and expended. Eating patterns, physical activity levels, sleep routine, genetics, among others, are main causes of the energy imbalance. According to the World Health Organization, one billion and nine hundred million adults worldwide were overweight in 2016; of these, 650 million were obese, corresponding to 39% and 13% of the world population, respectively. Some diseases are correlated with obesity, such as metabolic syndrome, insulin resistance, diabetes, cardiovascular diseases, musculoskeletal disorders, and some types of cancer. Obesity increases morbidity and mortality rates and reduces life quality and life expectancy. The images were obtained from Servier Medical Art 266 (http://smart.servier.com/, Accessed on December 21st 2022), licensed under a Creative 267 Commons Attribution 3.0 Unported License.

Figure 3: Schema of the adipogenesis in silico analysis pipeline. The computational simulation process of adipocyte development can be divided and implemented in an analysis pipeline. (1) The first step of the process is the collection of data on adipogenesis, including information about genes, proteins, and molecular mechanisms involved in the process. Data can be obtained from public databases, such as Gene Expression Omnibus (GEO), ArrayExpress, and Sequence Read Archive (SRA), or from experimental approaches, such as in vitro and in vivo assays, or clinical trials, among others. (2) The collected data are analyzed to identify the genes and proteins involved in adipogenesis, the molecular mechanisms involved in the process and the regulatory factors, and which ones are up and down-regulated. During the gene expression analysis step, Gene Ontology (GO) and Kyoto Encyclopedia of Genes and Genomes (KEEG) can be used to identify genes being expressed under different conditions and understand how these genes may be involved in adipogenesis processes. (3) After selecting the main factors, the hub genes are identified through the analysis of interactions between the factors: protein-protein interaction, target gene - miRNA interaction, or target gene - transcription factor (TF) interaction, among others. (4) Those with the highest degree of connections should preferably be selected for in vitro or in vivo experimental validation, which may use RT-PCR, gene silencing, or other techniques. (5) Finally, the main selected factors can be used from basic research to develop new drugs through 3D modeling methods, both the target (gene or protein) and small therapeutic molecules. The images were obtained and adapted from Servier Medical Art 266 (http://smart.servier.com/, Accessed on December 22st 2022), licensed under a Creative 267 Commons Attribution 3.0 Unported License.

Point 5: English needs minor editing.

Response 1: We thank the reviewer for the comment. We are performing additional revisions in the text and, as soon as they are finished, we will review the manuscript for English correction again. We have applied Grammarly for Microsoft® Office version 6.8.261.

General Comments:

We also have made some updates to the manuscript regarding the spellings of the corresponding author

From Alessandra Melo De Aguiar to Alessandra Melo de Aguiar

And we also included some more information in the figure caption, below (highlighted in red):

Figure 5: MSCs as an alternative method to test adipogenesis-related drugs in obesity research. The MSCs, isolated from several sources as adipose tissues, can be used as in vitro models to evaluate the pro-adipogenic or anti-obesity potential in drug discovery. Bidimensional culture represents the most employed in vitro method, although the least representative in terms of physiological conditions. To overcome this, 3D culture methods such as spheroid formation, association of cells to biomaterials as hydrogels and bio-printing are approaches currently used to improve cell-cell and cell-extracellular matrix interactions, resembling the human cells environment. A more complex and biologically-relevant model to mimic physiological functions of tissues or organs is the organ-on-a-chip technology, where cells are cultivated in 2D or 3D in microdevices, under a controlled microenvironment. In silico analysis assists all the in vitro stages with bioinformatic tools: databases and enrichment analysis softwares allow the analysis of high throughput data, while molecular docking and drug repurposing help in the discovery of drugs to specific targets. The integrated use of these alternative methods allows the translation from in vitro to in vivo tests using a reduced number of animals in the search for new AOM, also providing more information that can support clinical trials. AD-MSCs (Lonza®, Walkersville, USA; catalog number PT-5006) were cultivated for 14 days in the absence (undifferentiated MSCs) or presence (adipocytes) of adipogenic inducers and differentiation medium. Lipid droplet formation (green – Nile Red) and nuclei (blue – DAPI) images were obtained with a high-content imaging system (Operetta CLS®, PerkinElmer®, Waltham, MA, USA). For the schematic representation, the images were obtained from Servier Medical Art (http://smart.servier.com/, Accessed on 11/12/2022), licensed under a Creative Commons Attribution 3.0 Unported License.

We have added the term “and adapted” to all the figures made with images from Servier Medical Art, once we have made some adaptations to the original images.
